# Sensitive remote homology search by local alignment of small positional embeddings from protein language models

**Sean R Johnson\*, Meghana Peshwa, Zhiyi Sun\***

New England Biolabs Inc, Ipswich, United States

**Abstract** Accurately detecting distant evolutionary relationships between proteins remains an ongoing challenge in bioinformatics. Search methods based on primary sequence struggle to accurately detect homology between sequences with less than 20% amino acid identity. Profile- and structure-based strategies extend sensitive search capabilities into this twilight zone of sequence similarity but require slow pre-processing steps. Recently, whole-protein and positional embeddings from deep neural networks have shown promise for providing sensitive sequence comparison and annotation at long evolutionary distances. Embeddings are generally faster to compute than profiles and predicted structures but still suffer several drawbacks related to the ability of whole-protein embeddings to discriminate domain-level homology, and the database size and search speed of methods using positional embeddings. In this work, we show that low-dimensionality positional embeddings can be used directly in speed-optimized local search algorithms. As a proof of concept, we use the ESM2 3B model to convert primary sequences directly into the 3D interaction (3Di) alphabet or amino acid profiles and use these embeddings as input to the highly optimized Fold-seek, HMMER3, and HH-suite search algorithms. Our results suggest that positional embeddings as small as a single byte can provide sufficient information for dramatically improved sensitivity over amino acid sequence searches without sacrificing search speed.

**\*For correspondence:**
sjohnson@neb.com (SRJ);
sunz@neb.com (ZS)

## eLife assessment

This **important** study addresses the problem of detecting weak similarity between protein sequences, a procedure commonly used to infer homology or assign putative functions to uncharacterized proteins. The authors present a **convincing** approach that combines recently developed protein language models with well-established methods. The benchmarks provided show that the proposed tool is fast and accurate for remote homology detection, making this paper of general interest to all researchers working in the fields of protein evolution and genome annotation.

## Introduction

A common method for assigning a putative function to a protein sequence is to find sequences with experimentally determined functions that have similarities in sequence, structure, or evolutionary origin to the unannotated sequence (*Loewenstein et al., 2009*). Direct comparisons of primary sequence, for example using BLASTP (*Camacho et al., 2009*), are fast and reliable but show poor ability to detect homologs with less than about 20% identity to the query (*Rost, 1999*). Popular approaches for higher sensitivity sequence searches involve using sequence profiles, for example with PSI-BLAST (*Altschul et al., 1997*), HMMER3 (*Eddy, 2011*), HH-suite3 (*Steinegger et al., 2019*),

or MMseqs2 (*Steinegger and Söding, 2017*). Sequence profiles are derived from multiple sequence alignments (MSAs) and are often modeled as profile hidden Markov models (HMMs), for example in HMMER and HH-suite. Profile HMMs model each position as the probability of each amino acid at the position together with insertion and deletion probabilities. Because of their reliance on the construction of MSAs, profile-based methods can have high computational overhead for database construction, query preparation, or both.

Protein structure searches also show higher sensitivity than sequence searches (*Jambrich et al., 2023*). Until recently, the utility of structure searches for protein annotation was limited by the lack of extensive reference databases and the inability to predict structures quickly and reliably for sequences lacking experimentally determined structures. In the past several years, accurate protein structure prediction programs such as AlphaFold2 (*Jumper et al., 2021*) and ESMFold (*Lin et al., 2023*) have led to a massive increase in the size of databases of predicted protein structures. Coupled with fast structure search algorithms such as Foldseek (*van Kempen et al., 2024*), RUPEE (*Ayoub and Lee, 2019*), and Dali (*Holm, 2022*), structure prediction programs provide another powerful tool for remote homology detection. Foldseek achieves fast structure search by encoding the tertiary interactions of each amino acid in the 20-letter 3D interaction (3Di) alphabet. By using a structure alphabet of the same size as the amino acid alphabet, Foldseek can leverage optimized sequence search algorithms originally developed for amino acid sequences (*Steinegger and Söding, 2017*; *van Kempen et al., 2024*). Structure search methods suffer from some of the same drawbacks as profile-based methods, including the computational cost of converting primary sequences to predicted structures.

Emerging methods for protein annotation and remote homology detection rely on deep neural networks taking protein sequences as inputs and producing either a classification from a controlled vocabulary (*Bileschi et al., 2022*; *Sanderson et al., 2023*), a natural language description (*Gane et al., 2022*), positional embeddings, or a sequence embedding. Positional embeddings are fixed length vectors for each amino acid position of the protein. Positional embeddings produced by popular protein language models (pLMs) usually have large dimensions such as 1024 for ProtT5-XL-U50 (*Elnaggar et al., 2021*) and 2560 for ESM-2 3B (*Lin et al., 2023*). Sequence embeddings represent an entire sequence and are often calculated by element-wise averaging of the positional embeddings. Positional and sequence embeddings can be used for remote homology detection by using them to calculate substitution matrices in pairwise local alignments (*Kaminski et al., 2023*; *Pantolini et al., 2024*; *Ye and Iovino, 2023*), or by *k*-nearest neighbors searches (*Hamamsy et al., 2022*; *Schütze et al., 2022*), respectively.

While each of these emerging methods shows promise for improving sensitivity of protein search and annotation, they suffer various limitations. Classification models and methods relying on sequence embeddings struggle at discriminating individual domains of multi-domain proteins (*Bileschi et al., 2022*). Methods relying on large positional embeddings are space inefficient, and current search implementations are slow compared to other methods. Smaller positional embeddings would be more amenable to algorithmic optimizations using single instruction multiple data (SIMD) capabilities of central processing units (CPUs) that contribute to the speed of optimized sequence search algorithms (*Buchfink et al., 2021*; *Eddy, 2011*; *Kilinc et al., 2023*; *Steinegger et al., 2019*).

An embedding is an alternative vector representation of the input data which, preferably, makes the input data more suitable for some downstream task. We recognized that profile HMMs and 3Di sequences are types of protein positional embeddings with dimensionality as low as 1 (3Di sequences) to about 25 (profile HMMs, including amino acid frequencies and state transition probabilities), that are more suitable than amino acid sequences for the task of remote homology detection. We tested the hypothesis that the ESM-2 3B pLM (*Lin et al., 2023*) could be used to directly convert primary amino acid sequences into profile HMMs compatible with HMMER3 or HH-suite, and 3Di sequences compatible with Foldseek, providing a sequence search workflow leveraging the speed and sensitivity advantages of profile and structure search algorithms with an accelerated query preparation step enabled by the pLM.

## Results

### Using ESM-2 3B to generate small positional embeddings

ESM-2 was already pretrained on the masked language modeling task (*Devlin et al., 2019*; *Lin et al., 2023*) of predicting amino acid distributions at masked positions of input sequences, therefore no additional fine-tuning was necessary to induce it to produce probabilities compatible with profile HMM tools (*Figure 1A*). Positional amino acid frequencies predicted by ESM-2 3B resembled those found in MSAs built from sequence searches; *Figure 2* shows an example comparison of logos of HMM profiles (*Wheeler et al., 2014*) derived from the 4HBT Pfam profile and the YBGC_HELPY___14–90 sequence from the seed alignment. The example is cherry picked in terms of the profile length, to be short enough to look nice as a figure, but it is not cherry picked for agreement between the profiles; it was the first short one we looked at.

To produce Foldseek-compatible 3Di sequences, we trained a two-layer 1D convolutional neural network (CNN) to convert positional embeddings from the last transformer layer of ESM-2 3B into 3Di sequences, we then unfroze the last transformer layer and fine-tuned it together with the CNN. The fine-tuned model, which we call ESM-2 3B 3Di (*Figure 1B*), converted amino acids to 3Di with an accuracy of 64% compared to a test set of 3Di sequences derived from AlphaFold2-predicted structures. Training and test sets were derived from a random split of the Foldseek AlphaFold2 UniProt50 dataset (*Jumper et al., 2021*; *van Kempen et al., 2024*; *Varadi et al., 2022*), a reduced-redundancy subset of the UniProt AlphaFold2 structures (see Methods for details).

### Comparison of newly developed embedding methods

To evaluate the capacity of small embeddings to improve search sensitivity, we generated predicted profiles and 3Di sequences from clustered Pfam 32 splits (*Bileschi et al., 2022*) and converted them into formats compatible with various search tools (*Figure 1C*). Pfam (*Mistry et al., 2021*) is a set of manually curated MSAs of families of homologous protein domains. Some families presumed to have a common evolutionary origin are further grouped into clans. In the clustered splits, each family is divided into train and test groups such that each sequence in the test group has less than 25% identity to the most similar protein in the train group (*Bileschi et al., 2022*). The sensitivity of a search algorithm is evaluated by the ability to match sequences from the test groups to their corresponding train group at the family or clan level.

Methods using predicted profiles (*Figure 3A, B*) and those using predicted 3Di sequences (*Figure 3C, D*) both showed greater accuracy than phmmer (amino acid to amino acid) searches across all identity bins, and hmmscan (amino acid to profile HMM) searches on test sequences below 20% identity to the closest train sequence. Converting query sequences to hhsuite compatible profiles using ESM-2 3B and searching against databases built from ESM-2 predicted profiles of training sequences (*Figure 3A, B*, line 6) or profiles built from family-wise MSAs of the training sequences themselves (*Figure 3A, B*, line 5) gave improved accuracy at low identity bins compared to phmmer and hmmscan.

Foldseek searches where both the queries and database consisted of 3Di sequences produced by ESM-2 3B 3Di (*Figure 3C, D*, line 7) performed the best overall of all new methods tested. Foldseek considers both 3Di and amino acid sequences in its alignments and can therefore be conceptualized as using a 2-byte embedding. Running Foldseek in 3Di-only mode (line 8), a 1-byte embedding, led to a decrease in accuracy but still outperformed phmmer across all identity bins, and hmmscan on bins below 20% identity. We also tried creating HMMER3 profiles from predicted 3Di sequences (line 9). These performed worse than single-sequence Foldseek searches. Patching HMMER3 to use 3Di-derived background frequencies and Dirichlet priors (line 10) did not improve performance. Furthermore, we noticed that HMMER3 runs very slowly on 3Di sequences and profiles, presumably because the prefiltering steps were not optimized with 3Di sequences in mind.

While faster than MSA construction or full structure prediction, pLM embedding still has non-trivial computational overhead. This limits the possibility of using methods based on pLM embeddings to perform sensitive homology searches against large metagenomic databases such as the 2.4 billion sequence Mgnify database (*Richardson et al., 2023*). It would be desirable to have search methods where the database can remain as amino acid sequences or other cheaply calculated representations and the queries can be processed with more expensive methods. To this end, we tested hmmsearch using ESM-2 3B generated profiles as queries against amino acid databases (*Figure 3A and B*, lines

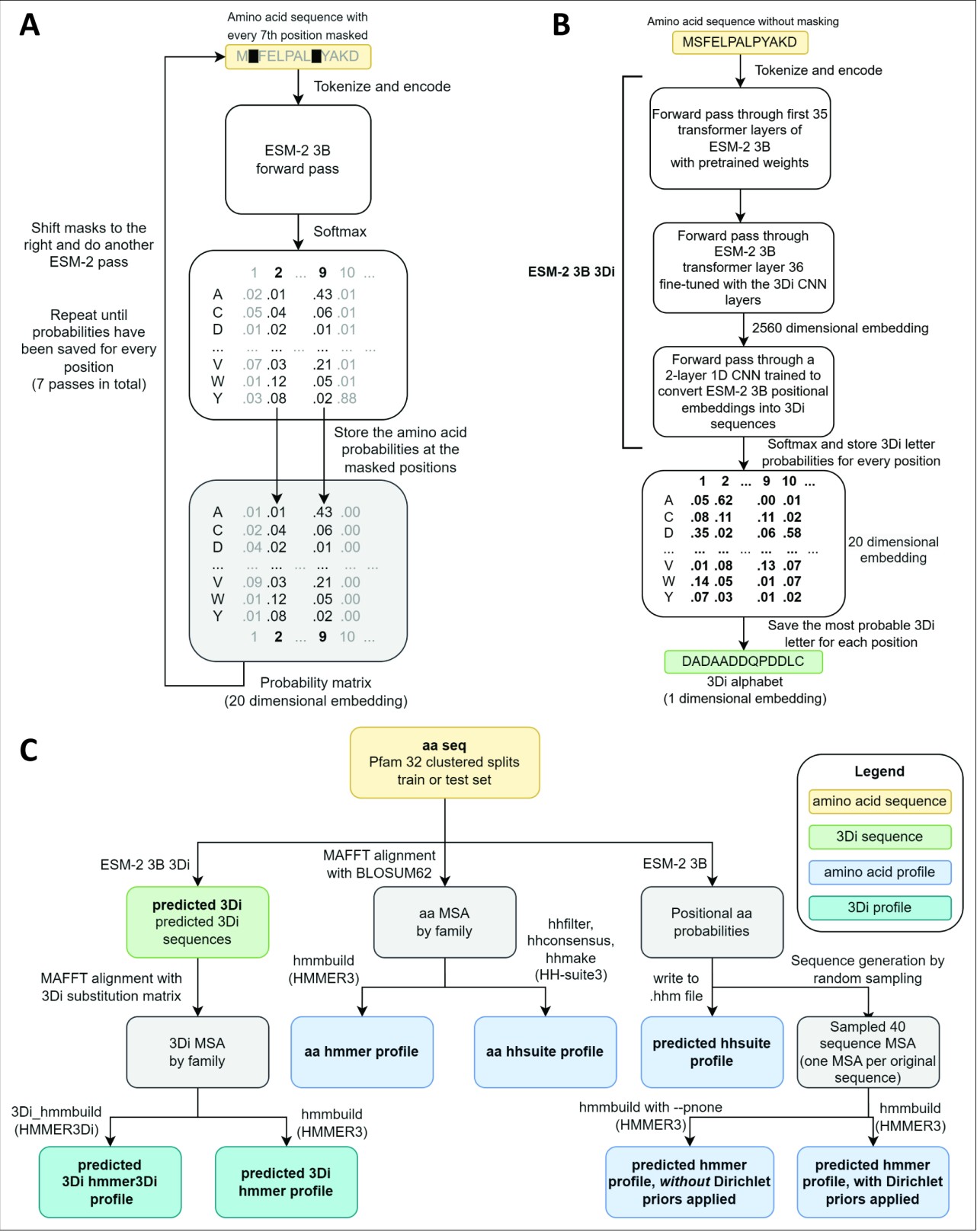

**Figure 1.** Schematics of embedding models and the experimental design. (**A**) ESM-2 3B can be directly used to predict amino acid probability distributions at masked positions. Our implementation uses seven passes. The second pass is shown in the figure. (**B**) ESM-2 3B 3Di, a fine-tuned ESM-2 3B with a small convolutional neural network (CNN) top model can be used to predict 3D interaction (3Di) sequences from amino acid sequences. (**C**) Data flow from amino acid sequences through embedding models and other programs to produce files used in homology searches. Bold words correspond to line labels in *Figure 3*.

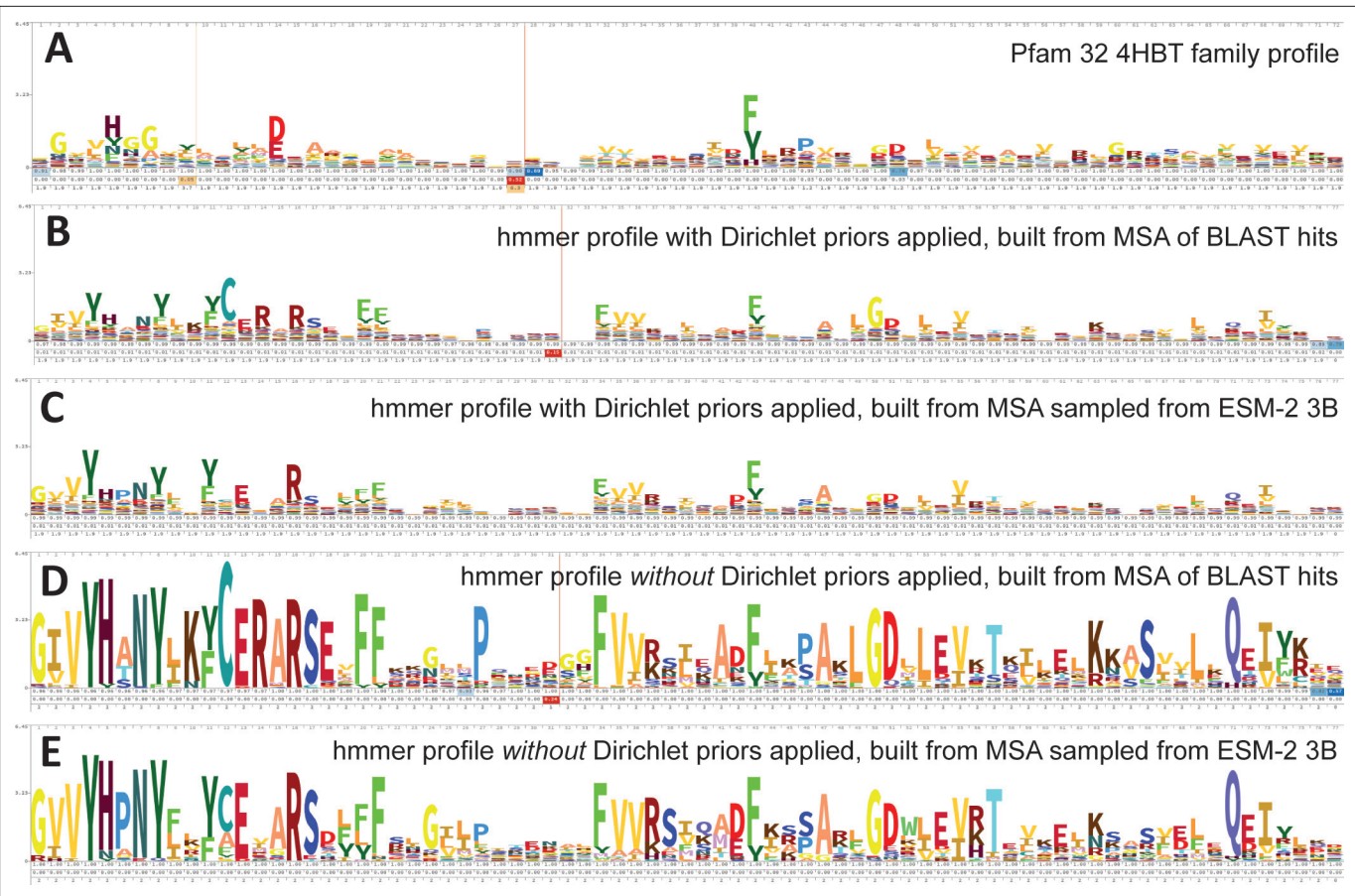

**Figure 2.** Logos related to the example test sequence YBGC_HELPY___14–90 from the 4HBT family. (**A**) 4HBT family hmm from Pfam 32. (**B**) hmmbuild with default settings on a multiple sequence alignment (MSA) of the top 100 hits supplied by an online blast search (https://blast.ncbi.nlm.nih.gov/Blast. cgi) of YBGC_HELPY___14–90 against the NCBI clustered nr database. (**C**) hmmbuild with default settings on the MSA sampled from the ESM-2 3B positional probabilities for YBGC_HELPY___14–90. (**D, E**) hmmbuild with Dirichlet priors disabled on the same MSAs as for (**B, C**), respectively. All logos were generated by uploading the corresponding.hmm file to https://skylign.org/ (**Wheeler et al., 2014**).

3 and 4). This is similar to a two-iteration PSI-BLAST (**Altschul et al., 1997**) or JackHMMER (**Johnson et al., 2010**) search where the first search and MSA-building step is replaced by a pLM embedding step. Curiously, hmmsearch using profiles built directly from the pLM probabilities (line 3) had the lowest accuracy of any algorithm. Nevertheless, profiles processed with hmmbuild from HMMER3, applying HMMER3 Dirichlet priors on top of the pLM probabilities (line 4), had better family prediction accuracy than phmmer at all but the highest identity bin, and accuracy on par with hmmscan at 18% identity and lower.

## Benchmarking of ESM-2 3B 3Di against emerging and established search methods

ESM-2 3B 3Di coupled to Foldseek search performed the best out of all the new methods we proposed on the clustered Pfam benchmark. To examine the performance of ESM-2 3B 3Di-based search against other emerging methods, we used the SCOPe40 benchmark (**Figure 4**; **Chandonia et al., 2019**; **van Kempen et al., 2024**), which has previously been used to evaluate Foldseek-based search methods (**Heinzinger et al., 2023**; **van Kempen et al., 2024**). The SCOPe40 benchmark is convenient because the dataset is much smaller than the clustered Pfam dataset, 11,211 vs 1,339,083 sequences. The small size of the dataset allowed us to compare compute-intensive methods, such as predicting AlphaFold2 structures for the entire dataset and then running Foldseek. In the SCOPe40 benchmark, the criteria for true positives (TPs) vary depending on the level of classification: for family, TPs are matches within the same family; for superfamily, TPs are matches sharing the same superfamily

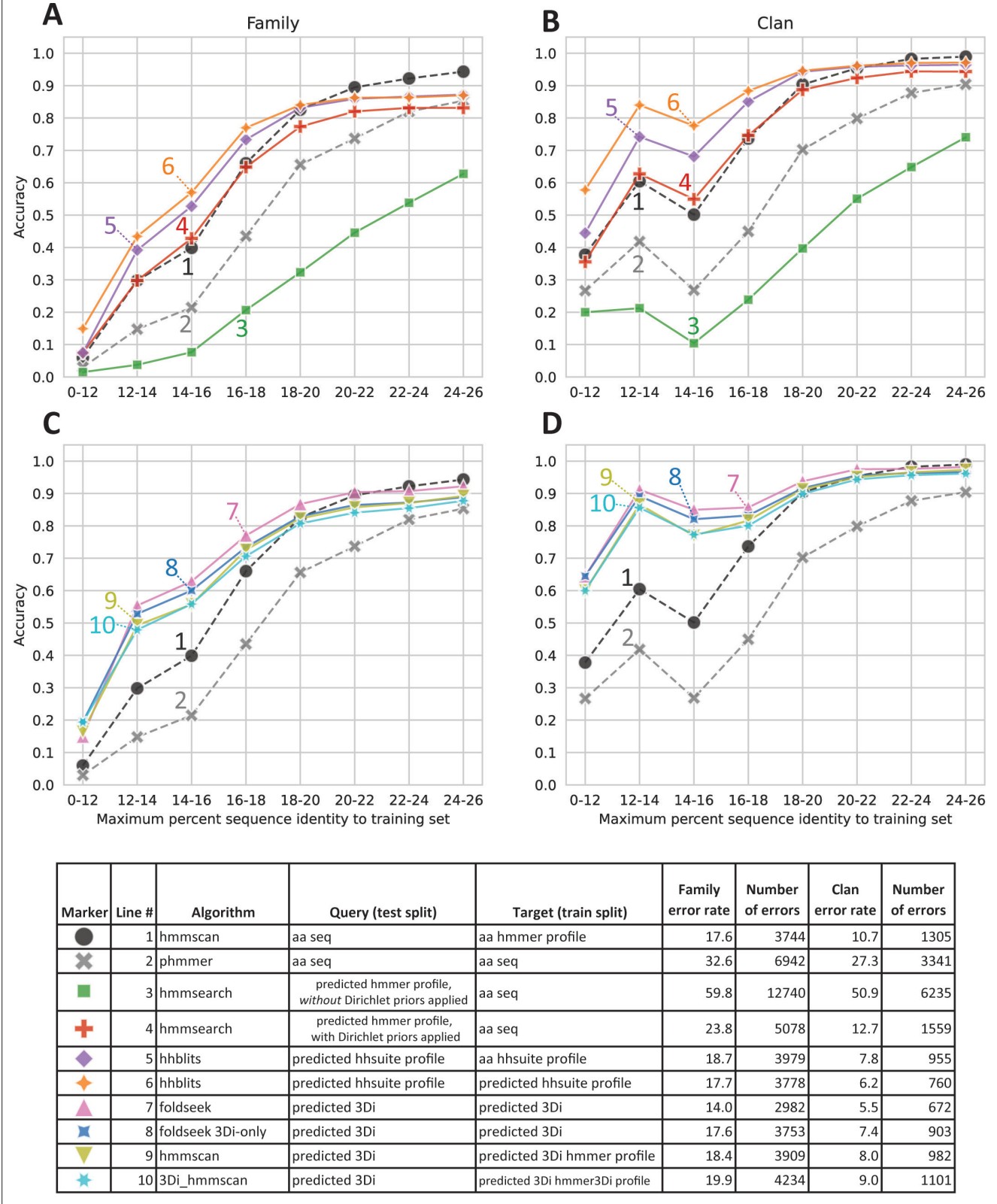

**Figure 3.** Homology detection accuracy. Test sequences were binned based on percent identity to the closest training sequence in the same family and annotated based on the top scoring hit from a search against the entire set of training sequences or training sequence family profiles, depending on the algorithm. (**A, C**) Family recovery accuracy by bin. (**B, D**) Clan recovery accuracy. (**A, B**) Compare amino acid profile-based methods. (**C, D**) Compare Foldseek-based methods. Dashed lines are controls. There are a total of 21,293 test sequences. 12,246 test sequences have clan assignments.

The table accompanying the figure:

| Marker | Line # | Algorithm | Query (test split) | Target (train split) | Family error rate | Number of errors | Clan error rate | Number of errors |
|---|---|---|---|---|---|---|---|---|
| ● | 1 | hmmscan | aa seq | aa hmmer profile | 17.6 | 3744 | 10.7 | 1305 |
| ✕ | 2 | phmmer | aa seq | aa seq | 32.6 | 6942 | 27.3 | 3341 |
| ■ | 3 | hmmsearch | predicted hmmer profile, *without* Dirichlet priors applied | aa seq | 59.8 | 12740 | 50.9 | 6235 |
| ✚ | 4 | hmmsearch | predicted hmmer profile, with Dirichlet priors applied | aa seq | 23.8 | 5078 | 12.7 | 1559 |
| ◆ | 5 | hhblits | predicted hhsuite profile | aa hhsuite profile | 18.7 | 3979 | 7.8 | 955 |
| ◆ | 6 | hhblits | predicted hhsuite profile | predicted hhsuite profile | 17.7 | 3778 | 6.2 | 760 |
| ▲ | 7 | foldseek | predicted 3Di | predicted 3Di | 14.0 | 2982 | 5.5 | 672 |
| ★ | 8 | foldseek 3Di-only | predicted 3Di | predicted 3Di | 17.6 | 3753 | 7.4 | 903 |
| ▼ | 9 | hmmscan | predicted 3Di | predicted 3Di hmmer profile | 18.4 | 3909 | 8.0 | 982 |
| ★ | 10 | 3Di_hmmscan | predicted 3Di | predicted 3Di hmmer3Di profile | 19.9 | 4234 | 9.0 | 1101 |

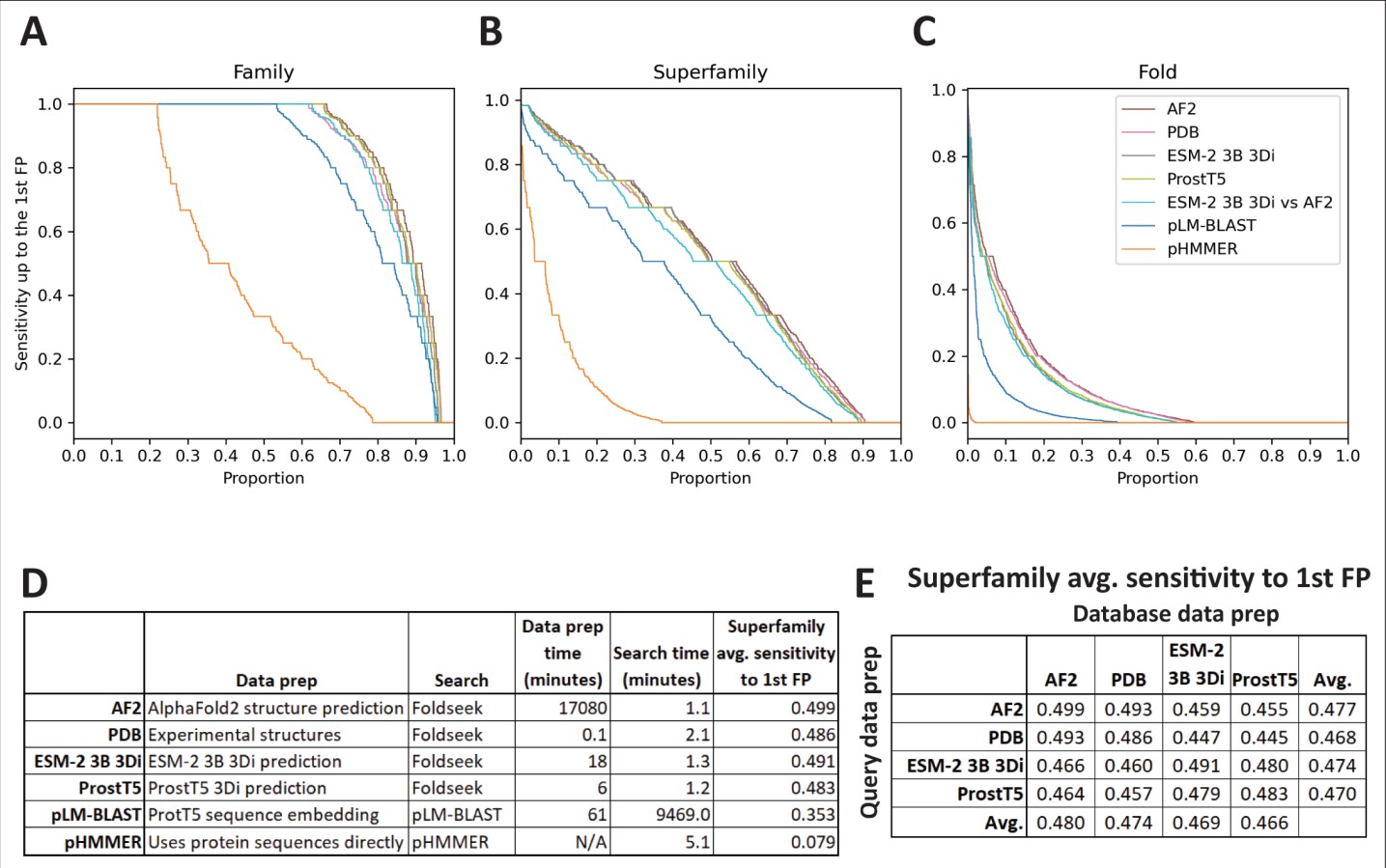

**Figure 4.** SCOPe40 benchmark. Cumulative distribution plots of the number of queries attaining each level of sensitivity to the first false positive fold at the (**A**) family, (**B**) superfamily, and (**C**) fold level. (**D**) Data preparation, search times, and average sensitivity at the superfamily level. (**E**) Comparison of average sensitivity at the superfamily level of Foldseek run with queries and databases prepared by different methods.

but not the same family; and for fold, TPs are matches sharing the same fold but are not the same superfamily. Any hits from different folds (but not from different families or superfamilies) are considered false positives (FPs). Calculating the metric in this way has the effect that 'Family' is a metric of homolog detection at the closest evolutionary distance (*Figure 4A*), 'Superfamily' is a metric of detection of more distant homologs (*Figure 4B*), and 'Fold' is a metric of very distant homolog detection or detection of evolutionarily unrelated proteins with similar folded structures (*Figure 4C*). All 11,211 proteins were used as queries for timing, but for sensitivity calculations, only the 3566 proteins with at least one other family member, at least one other superfamily member, and at least one other fold member were considered.

We benchmarked against Foldseek using experimentally determined structures, 3Di sequences predicted by ProstT5 (*Heinzinger et al., 2023*), and 3Di sequences derived from structures predicted by AlphaFold2 (*Jumper et al., 2021; Mirdita et al., 2022*). We also benchmarked a method based on local alignment of large positional embeddings, pLM-BLAST (*Kaminski et al., 2023*). Protein sequence search with pHMMER search was included as a control.

The Foldseek-based methods all performed similarly on the SCOPe40 benchmark at all three homology levels, all outperforming pLM-BLAST and far outperforming pHMMER (*Figure 4*). At the superfamily level, AlphaFold2-predicted structures provided the best average sensitivity. 3Di sequences generated by different methods could be used as queries against databases built from 3Di sequences generated by other methods, but with some degradation of sensitivity compared to searches where both query and database 3Di sequences were generated by the same method (*Figure 4E*).

For each method, we also timed the data preparation time and search time (*Figure 4D*). pHMMER uses protein sequences directly and does not require a database building step, so database and query preparation are not needed. For data preparation time, AlphaFold2 structure prediction was by far the slowest, taking nearly 2 weeks on our hardware (Methods), and ProstT5 was the fastest, converting all protein sequences to 3Di sequences in just 6 min. For search time, the Foldseek-based methods were the fastest. pLM-BLAST was much slower than Foldseek-based methods, which makes sense because it is doing the same types of Smith-Waterman alignments as Foldseek, but with positional encodings more than 500 times as large. Pre-filtering is used in both methods, but evidently the pre-filtering in pLM-BLAST is not enough to compensate for the slowness of the alignment step.

## Discussion

We tested several schemes for using the ESM-2 3B pLM to recode protein sequences into formats compatible with profile HMM or structure search tools with the hope of enhancing search sensitivity compared to primary sequence searches, but with less computational overhead than MSA construction or full structure prediction. The most successful of our efforts came from fine-tuning ESM-2 3B to convert protein sequence into 3Di sequences and running Foldseek structure search.

Compared to other emerging methods for neural-network-driven remote homology detection, ESM-2 3B 3Di coupled to Foldseek enabled faster and more sensitive search than pLM-BLAST (*Kaminski et al., 2023*), and was slower but had similar sensitivity to ProstT5 (*Heinzinger et al., 2023*) coupled to Foldseek. Furthermore, 3Di sequences predicted by ESM-2 3B 3Di or by ProstT5 seem to be compatible with Foldseek searches against experimental or AlphaFold2-generated structures (*Figure 4E*), enabling a sequence annotation workflow where large numbers of proteins, for example from a newly sequenced genome, can be quickly annotated with high sensitivity by pLM-based conversion to 3Di sequences followed by Foldseek search against existing databases of experimental or predicted structures, for example the protein data bank (*Berman et al., 2000*) or the AlphaFold Protein Structure Database (*Varadi et al., 2022*).

There are many possible directions for future development of improved embeddings, faster conversion and search programs, and comprehensive reference databases. It is significant that a 1-byte embedding, the predicted 3Di sequences run in 3Di-only Foldseek searches, was among the top performing models. This result suggests that positional embeddings as small as a single byte can provide sufficient information for dramatically improved sensitivity over amino acid sequence searches. The 3Di alphabet was not developed to maximize remote homology search sensitivity, but to model tertiary interactions in protein structures. In future work, small positional embeddings optimal for local alignment could be learned directly from a differentiable alignment algorithm (*Petti et al., 2023*) instead of relying on proxy tasks of amino acid or 3Di prediction.

Finally, asymmetric architectures where embedding a database sequence is cheaper than embedding a query, analogous to searches with profile HMM queries against primary sequence databases, could be a powerful method for improving search sensitivity against large and growing reference databases. Some first steps in that direction could be conversion to profile HMM-style embeddings in a single pass, rather than the seven we required, and predicting state transition probabilities, which may lead to improved search performance. We hazzve made our model training, search, and data analysis code publicly available. We hope our results and code will serve as a springboard for further exploration of the utility of low-dimensionality positional embeddings of protein sequences.

## Methods
### Alignments

Unless otherwise noted, MSAs were made using MAFFT (v7.505) (*Katoh and Standley, 2013*) with options `--anysymbol --maxiterate` 1000 `--globalpair`. Protein alignments used the BLOSUM62 substitution matrix (*Henikoff and Henikoff, 1992*). 3Di alignments used the 3Di substitution matrix from Foldseek (*van Kempen et al., 2024*; *Steinegger Lab, 2022*; https://github.com/steineggerlab/foldseek/blob/master/data/mat3di.out).

## Patching HMMER3 with background frequencies and Dirichlet priors for 3Di

We created a fork of the HMMER3 program (*Eddy, 2011*), replacing amino acid background frequencies and Dirichlet priors with values calculated from the 3Di alphabet instead of the amino acid alphabet (GitHub, copy archived at *Johnson, 2024b*; https://github.com/seanrjohnson/hmmer3di/commit/347ee45cd1fb5fc5984b2a3e3e188afbde4b8d2f). To generate a set of 3Di MSAs, we converted the AlphaFold UniProt Foldseek database (*Jumper et al., 2021*; *van Kempen et al., 2024*; *Varadi et al., 2022*) to a 3Di fasta file. We then looked up every sequence name from the Pfam 35 seed file in the UniProt 3Di fasta file and, for cases where the corresponding sequence was identifiable, extracted the sub-sequence corresponding to the Pfam 35 seed. 3Di seeds from each profile were aligned using MAFFT. MSA columns with more than 10 rows were used to calculate background frequencies and Dirichlet priors using the HMMER3 program esl-mixdchlet fit with options -s 17 9 20. Amino acid background frequencies and Dirichlet priors in the HMMER3 source code were then replaced with the newly calculated 3Di background frequencies and Dirichlet priors. We call the patched HMMER3 as HMMER3Di.

## Fine-tuning ESM-2 3B to convert amino acid sequences into 3Di sequences

A 1D CNN was added on top of ESM-2 3B. The CNN takes as input position-wise embeddings from the last transformer layer of ESM-2 3B. The CNN consists of two layers, the first layer has 2560 input channels (the size of the embeddings from ESM-2 3B), and 300 output channels, kernel size 5, stride 1, padding 3. The second layer has 300 input channels and 21 output channels (one for each 3Di symbol plus a padding symbol), kernel size 5, stride 1, padding 1. The model was trained with a weighted cross-entropy loss function using weights of 0.1 * the diagonal from 3Di substitution matrix. The neural network was implemented in PyTorch (*Paszke et al., 2019*).

Training data were derived from the Foldseek AlphaFold2 UniProt50 dataset (*Jumper et al., 2021*; *van Kempen et al., 2024*; *Varadi et al., 2022*), a reduced-redundancy subset of the UniProt AlphaFold2 structures. The Foldseek database was downloaded using the Foldseek 'databases' command line program, converted into protein and 3Di fasta files, filtered to remove sequences smaller than 120 amino acids and larger than 1000 amino acids, and split into train, validation, and test subsets, 90%:5%:5% (33,924,764:1,884,709:1,884,710 sequences).

With ESM-2 layers frozen, the CNN was trained on the task of converting amino acids to 3Di sequences using the AdamW optimizer with learning rate 0.001, weight decay 0.001, and exponential learning rate decrease (gamma 0.98, applied every 100 batches). Training sequences were randomly selected in batches of 15 sequences. Training proceeded for 1301 batches, leading to a training accuracy of about 58%.

The last transformer layer of ESM-2 was then unfrozen and training restarted from the saved weights, with the same training parameters. Training continued for another 24,001 batches of 10 random training sequences. Accuracy on the final training batch was 65%. Using the final trained weights, test sequences were converted to 3Di at an accuracy of 64.4%. We call the fine-tuned model ESM-2 3B 3Di.

The trained weights are available on Zenodo. The training code is available on GitHub (copy archived at *Johnson, 2024a*). The model training code should be useful for fine-tuning ESM-2 to convert amino acid sequences to various other kinds of sequences, such as secondary structure codes.

## Pfam 32 clustered splits

Pfam 32 clustered splits (*Bileschi et al., 2022*) were downloaded from: https://console.cloud.google.com/storage/browser/brain-genomics-public/research/proteins/pfam/clustered_split. Data for mapping of individual Pfam 32 families to clans were downloaded from: https://ftp.ebi.ac.uk/pub/databases/Pfam/releases/Pfam32.0/. The sequences from the train, dev, and test splits were sorted into unaligned fasta files according to their split and family (**aa seq**).

## Predicting 3Di sequences

Each training and test sequence was converted into a predicted 3Di sequence (predicted 3Di) using the ESM-2 3B 3Di model described above. MAFFT alignments were made of both the amino acid

and predicted 3Di training and test sequences for each family. HMMER3 profiles were built from the alignments using either unaltered HMMER3 (predicted 3Di hmmer profile), or HMMER3Di (predicted 3Di hmmer3Di profile).

## Predicting profiles to generate HH-suite hhm files and HMMER3 hmm files from single sequences

Positional amino acid probabilities were calculated for unaligned train and test sequence using pretrained ESM-2 3B in the following algorithm.

1. Prepend sequence with M. This is because ESM-2 3B has a strong bias toward predicting M as the first amino acid in every sequence, and most Pfam domains do not start with M.
2. Mask the first, eighth, etc. position in the sequence
3. Run a forward pass of ESM-2 3B over the masked sequence.
4. Save the logits for each of the 20 amino acid tokens for each masked position.
5. Shift the masks one position to the right.
6. Repeat steps 3 through 5 another six times until logits have been saved for every position.
7. Use the softmax function to calculate the amino acid probabilities at each position from the logits.

Note that in our actual implementation, a single forward pass was run on a batch of seven copies of the input sequence, each with different masking.

The hyperparameter of seven passes over the input sequence was chosen semi-empirically. Ideally, masking would be done one position at a time, such that each position benefits from the context of the rest of the sequence. In practice, masking individual positions is prohibitively slow. We experimented with different masking distances in a non-systematic way and found that probabilities derived from seven passes gave similar probabilities as masking each position individually. In addition, in the original ESM-2 pretraining, 15%, approximately 1/7, of positions were masked at each training step, so our masking of every 7th position resembles the training conditions.

The positional probabilities were written directly as an HH-suite compatible.hhm file (predicted hhsuite profile). A 40 sequence fasta MSA file was written where each sequence was randomly sampled from the probability distribution. Hmmbuild was run with default settings on the sampled MSA (predicted hmmer profile, with Dirichlet priors applied) and with the -pn;on; setting, which disables adjustments to the probability distribution based on Dirichlet priors (predicted hmmer profile, *without* Dirichlet priors applied).

## Building HH-suite hhm and HMMER3 hmm profiles from train amino acid MSAs

The amino acid MSA for each training family were converted into an HH-suite database (**aa hhsuite profile**) with the following bash script:

```
echo '#'$profile_name >msa/${profile_name}.a3m
hhfilter -i $MSA_FASTA -a msa/${profile_name}.a3m -M 50 -id 90;
hhconsensus -i msa/${profile_name}.a3m -o consensus/${profile_name}.a3m
hhmake -name $base -i consensus/${profile_name}.a3m -o hhm/${base}.hhm
ffindex_build -s db_hhm.ffdata db_hhm.ffindex hhm
ffindex_build -s db_a3m.ffdata db_a3m.ffindex consensus
cstranslate -f -x 0.3 c 4 -I a3m -i db_a3m -o db_cs219
```

HMMER3 profiles were built by calling hmmbuild with default settings on the training family amino acid MSAs (**aa hmmer profile**).

## Building HH-suite databases from predicted profiles

HH-suite databases were built from predicted profiles.hhm files and the corresponding sampled 40 sequence MSA fasta files using the following bash script:

```
ffindex_build -s db_hhm.ffdata db_hhm.ffindex esm2_3B_profiles
ffindex_build -s db_a3m.ffdata db_a3m.ffindex esm2_3B_sampled_msascstranslate
-f -x 0.3 c 4 -I a3m -i db_a3m -o db_cs219
```

## Building Foldseek databases from predicted 3Di sequences

Amino acid and predicted 3Di fasta files were converted into Foldseek-compatible databases using a new script, fasta2foldseek.py, available from the esmologs python package (see below).

## Hmmscan, phmmer, and hmmsearch HMMER3 searches

In an attempt to mimic the Top pick HMM strategy reported by *Bileschi et al., 2022*, we ran all HMMER3 searches in up to two iterations. The first iteration was run with default settings. For test sequences where no hits were detected among the training sequences or profiles, depending on the program, a second iteration was run with the addition of parameters intended to maximize sensitivity at the expense of search speed:

```
--max -Z 1 --domZ 1 -E 1000000 --domE 1000000
```

It should be noted that while our phmmer results are directly comparable to the phmmer results reported by Bileschi et al., our hmmscan results are not directly comparable to the reported 'Top Pick HMM' results because we re-aligned the training sequences for each family instead of using the Pfam seed alignments. Still our results were very similar. We observed a 17.6% error rate (3744 test sequences with mispredicted family assignments) by hmmscan, compared to the reported 18.1% error rate (3844 mispredictions).

## 3Di_hmmscan HMMER3Di searches

3Di_hmmscan searches were performed using the same two-iteration method described above for searches using standard HMMER3 programs.

## Hhblits HH-suite searches

Hhbilts was run with the options:

```
-tags -n 1 v 0
```

## Foldseek searches

After converting both query and target amino acid and predicted 3Di fasta files into Foldseek-compatible databases (see above), Foldseek searches were run with the following commands:

```
foldseek search test_db train_db foldseek_results tmpFolder
foldseek convertalis test_db train_db foldseek_results hits.tsv --format-
output query,target,bits
```

For 3Di-only searches, the option `--alignment-type` 0 was added to the search call.

## SCOPe40 benchmark

The SCOPe40 benchmark was conducted as previously described (*Heinzinger et al., 2023*; *van Kempen et al., 2024*). For timings, GPU operations were run on an Nvidia A100 GPU with 40 Gb of VRAM. CPU operations were run on 16 cores on an Intel Xeon Gold 6258 R.

AlphaFold2 structures were computed using local ColabFold (*Jumper et al., 2021*; *Mirdita et al., 2022*), with the command:

```
colabfold_batch -num-recycle 3 -num-models 5 peptides.fasta colabfold
```

ProstT5 embedding was run with default settings of the script: https://github.com/mheinzinger/ProstT5/blob/f93a7a1b696a74acee9ce85226ba9047d74f96fe/scripts/predict_3Di_encoderOnly.py (*Heinzinger et al., 2023*; *Heinzinger, 2023*).

Foldseek was always run with the command:

```
Foldseek search queryDB targetDB aln3 tmpFolder -s 9.5 -e 10 --max-seqs
2000 --threads 16
```

## pLM-BLAST for SCOPe40

For pLM-BLAST (*Kaminski et al., 2023*), the version from October 30, 2023 was used (*Dunin-Horkawicz et al., 2023*) . We had to make a few trivial changes to the code to make it not crash when running the following operations.

The pLM-BLAST database was built with the entire SCOPe40 protein set (11,211 sequences), using the commands:

```
python pLM-BLAST/embeddings.py start pLM-blastDB.fasta pLM-blastDB
-embedder pt --gpu -bs 0 --asdir -t 1500
python pLM-BLAST/scripts/dbtofile.py pLM-blastDB
```

For the sake of speed, for embedding and searching queries, the SCOPe40 was divided into 10 equal partitions. Each partition was converted to embeddings with the command:

```
python pLM-BLAST/embeddings.py start peptides_partition_[num].fasta
peptides_partition_[num].pt --gpu -bs 0 t 1500
```

Timings for database prep were the sum of timings for all the embedding calls.
Each query partition was searched against the database with the command:

```
python pLM-BLAST/scripts/plmblast.py pLM-blastDB peptides_partition_[num]
[num]_hits.csv --use_chunks -workers 16
```

Timings for the search were the sum of the timings of all of the plmblast.py calls.

## Acknowledgements

We thank Sergey Ovchinnikov for helpful discussion about protein language models, Sean Eddy for helpful discussion about HMMER3, and Gary Smith for IT support.

## Additional information

### Competing interests

Sean R Johnson, Meghana Peshwa, Zhiyi Sun: Employee of New England Biolabs Inc.

### Funding

| Funder | Grant reference number | Author |
|---|---|---|
| New England Biolabs | | Sean R Johnson<br>Meghana Peshwa<br>Zhiyi Sun |

The funders had no role in study design, data collection, and interpretation, or the decision to submit the work for publication.

### Author contributions

Sean R Johnson, Conceptualization, Software, Investigation, Visualization, Methodology, Writing – original draft; Meghana Peshwa, Data curation, Software, Investigation, Writing – review and editing; Zhiyi Sun, Supervision, Methodology, Project administration, Writing – review and editing

### Author ORCIDs

Sean R Johnson (ID) https://orcid.org/0000-0001-8261-9015

Reviewer #1 (Public Review): https://doi.org/10.7554/eLife.91415.3.sa1
Author Response https://doi.org/10.7554/eLife.91415.3.sa2

## Additional files

### Supplementary files
• MDAR checklist

### Data availability
HMMER3 patched with 3Di background frequencies and Dirichlet priors: https://github.com/seanr-johnson/hmmer3di; copy archived at *Johnson, 2024b*. Code for neural network training, sequence searches, and data analysis: https://github.com/seanrjohnson/esmologs; copy archived at *Johnson, 2024a*, Model weights for ESM-2 3B 3Di, predicted profiles and 3Di sequences from the Pfam 32 clustered splits, and other data necessary to reproduce the analyses: https://doi.org/10.5281/zenodo.8174959.

The following dataset was generated:

| Author(s) | Year | Dataset title | Dataset URL | Database and Identifier |
|---|---|---|---|---|
| Johnson SR, Peshwa M, Sun Z | 2023 | Sensitive remote homology search by local alignment of small positional embeddings from protein language models | https://doi.org/10.5281/zenodo.8174959 | Zenodo, 10.5281/zenodo.8174959 |

The following previously published datasets were used:

| Author(s) | Year | Dataset title | Dataset URL | Database and Identifier |
|---|---|---|---|---|
| Varadi M, Anyango S, Deshpande M, Nair S, Natassia C, Yordanova G, Yuan D, Stroe O, Wood G, Laydon A, Žídek A, Green T, Tunyasuvunakool K, Petersen S, Jumper J, Clancy E, Green R, Vora A, Lutfi M, Figurnov M, Cowie A, Hobbs N, Kohli P, Kleywegt G, Birney E, Hassabis D, Velankar S | 2023 | AlphaFold2 UniProt50 | https://foldseek.steineggerlab.workers.dev/ | foldseek, V4 |
| Chandonia J-M, Fox NK, Brenner SE | 2012 | SCOPe 2.01 | https://wwwuser.gwdg.de/~compbiol/foldseek/scop40pdb.tar.gz | SCOPe, 2.01 |
| Bileschi ML, Belanger D, Bryant DH, Sanderson T, Carter B, Sculley D, Bateman A, DePristo MA, Colwell LJ | 2022 | Pfam 32 clustered splits | https://console.cloud.google.com/storage/browser/brain-genomics-public/research/proteins/pfam/clustered_split | Pfam, v32 |

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
