## [Editor Report · eLife assessment]

This **important** study addresses the problem of detecting weak similarity between protein sequences, a procedure commonly used to infer homology or assign putative functions to uncharacterized proteins. The authors present a **convincing** approach that combines recently developed protein language models with well-established methods. The benchmarks provided show that the proposed tool is fast and accurate for remote homology detection, making this paper of general interest to all researchers working in the fields of protein evolution and genome annotation.

---

## [Referee Report · Reviewer #1 (Public Review)]

This paper describes a new method for sequence-based remote homology detection. Such methods are essential for the annotation of uncharacterized proteins and for studies of protein evolution.

The main strength and novelty of the proposed approach lies in the idea of combining state-of-the-art sequence-based (HHpred and HMMER) and structure-based (Foldseek) homology detection methods with protein language models (the ESM2 model was used). The authors show that high-dimensional, information-rich representations extracted from the ESM2 model can be efficiently combined with the aforementioned tools.

The benchmarking of the new approach is convincing and shows that it is suitable for homology detection at very low sequence similarity. The method is also fast because it does not require the computation of multiple sequence alignments for profile calculation or structure prediction.

Overall, this is an interesting and useful paper that proposes an alternative direction for the problem of distant homology detection.

---

## [Author Response]

The following is the authors’ response to the original reviews.

**Public Reviews:**

**Reviewer #1 (Public Review):**
Summary:This work describes a new method for sequence-based remote homology detection. Such methods are essential for the annotation of uncharacterized proteins and for studies of protein evolution.Strengths:The main strength and novelty of the proposed approach lies in the idea of combining stateof-the-art sequence-based (HHpred and HMMER) and structure-based (Foldseek) homology detection methods with recent developments in the field of protein language models (the ESM2 model was used). The authors show that features extracted from high-dimensional, information-rich ESM2 sequence embeddings can be suitable for efficient use with the aforementioned tools.The reduced features take the form of amino acid occurrence probability matrices estimated from ESM2 masked-token predictions, or structural descriptors predicted by a modified variant of the ESM2 model. However, we believe that these should not be called "embeddings" or "representations". This is because they don't come directly from any layer of these networks, but rather from their final predictions.

We agree that there is some room for discussion about whether the amino acid probabilities returned by pre-trained ESM-2 and the 3Di sequences returned by ESM-2 3B 3Di can be properly referred to as “embeddings”. The term “embedding” doesn’t have a formal definition, other than some kind of alternative vector representation of the input data which, preferably, makes the input data more suitable for some downstream task. In that simple sense of the word “embedding”, amino acid probabilities and 3Di sequences output by our models are, indeed, types of embeddings. We posed the question on Twitter(https://twitter.com/TrichomeDoctor/status/1715051012162220340) and nobody responded, so we are left to conclude that the community is largely ambivalent about the precise definition of “embedding”.

We’ve added language in our introduction to make it more clear that this is our working definition of an “embedding”, and why that definition can apply to profile HMMs and 3Di sequences.

The benchmarks presented suggest that the approach improves sensitivity even at very low sequence identities <20%. The method is also expected to be faster because it does not require the computation of multiple sequence alignments (MSAs) for profile calculation or structure prediction.Weaknesses:The benchmarking of the method is very limited and lacks comparison with other methods. Without additional benchmarks, it is impossible to say whether the proposed approach really allows remote homology detection and how much improvement the discussed method brings over tools that are currently considered state-of-the-art.

We thank the reviewer for the comment. To address the question, we’ve expanded the results by adding a new benchmark and added a new figure, Figure 4. In this new content, we use the SCOPe40 benchmark, originally proposed in the Foldseek paper (van Kempen et al., 2023), to compare our best method, ESM-2 3B 3Di coupled to Foldseek, with several other recent methods. We find our method to be competitive with the other methods.

We are hesitant to claim that any of our proposed methods are state-of-the-art because of the lack of a widely accepted standard benchmark for remote homology detection, and because of the rapid pace of advancement of the field in recent years, with many groups finding innovative uses of pLMs and other neural-network models for protein annotation and homology detection.

**Reviewer #2 (Public Review):**
Summary:The authors present a number of exploratory applications of current protein representations for remote homology search. They first fine-tune a language model to predict structural alphabets from sequence and demonstrate using these predicted structural alphabets for fast remote homology search both on their own and by building HMM profiles from them. They also demonstrate the use of residue-level language model amino acid predicted probabilities to build HMM profiles. These three implementations are compared to traditional profile-based remote homology search.Strengths:Predicting structural alphabets from a sequence is novel and valuable, with another approach (ProstT5) also released in the same time frame further demonstrating its application for the remote homology search task.Using these new representations in established and battle-tested workflows such as MMSeqs, HMMER, and HHBlits is a great way to allow researchers to have access to the state-of-the-art methods for their task.Given the exponential growth of data in a number of protein resources, approaches thatallow for the preparation of searchable datasets and enable fast search is of high relevance.Weaknesses:The authors fine-tuned ESM-2 3B to predict 3Di sequences and presented the fine-tuned model ESM-2 3B 3Di with a claimed accuracy of 64% compared to a test set of 3Di sequences derived from AlphaFold2 predicted structures. However, the description of this test set is missing, and I would expect repeating some of the benchmarking efforts described in the Foldseek manuscript as this accuracy value is hard to interpret on its own.

The preparation of training and test sets are described in the methods under the heading “Fine tuning ESM-2 3B to convert amino acid sequences into 3Di sequences”. Furthermore, there is code in our github repository to reproduce the splits, and the entire model training process:https://github.com/seanrjohnson/esmologs#train-esm-2-3b-3di-starting-from-the-esm-2-3bpre-trained-weights

We didn’t include the training/validation/test splits in the Zenodo repository because they are very large: train 33,924,764; validation 1,884,709; test 1,884,710 sequences, times 2 because there are both amino acid and 3Di sequences. It comes out to about 30 Gb total, and is easily rebuilt from the same sources we built it from.

We’ve added the following sentence to the main text to clarify:

“Training and test sets were derived from a random split of the Foldseek AlphaFold2 UniProt50 dataset (Jumper et al., 2021; van Kempen et al., 2023; Varadi et al., 2022), a reducedredundancy subset of the UniProt AlphaFold2 structures (see Methods for details).”

To address the concern about comparing to Foldseek using the same benchmark, we’ve expanded the results section and added a new figure, Figure 4 using the SCOPe40 benchmark originally presented in the Foldseek paper, and subsequently in the ProstT5 paper to compare Foldseek with ESM-2 3B 3Di to Foldseek with ProstT5, AlphaFold2, and experimental structures.

Given the availability of predicted structure data in AFDB, I would expect to see a comparison between the searches of predicted 3Di sequences and the "true" 3Di sequences derived from these predicted structures. This comparison would substantiate the innovation claimed in the manuscript, demonstrating the potential of conducting new searches solely based on sequence data on a structural database.

See response above. We’ve now benchmarked against both ProstT5 and AF2.

The profile HMMs built from predicted 3Di appear to perform sub-optimally, and those from the ESM-2 3B predicted probabilities also don't seem to improve traditional HMM results significantly. The HHBlits results depicted in lines 5 and 6 in the figure are not discussed at all, and a comparison with traditional HHBlits is missing. With these results and presentation, the advantages of pLM profile-based searches are not clear, and more justification over traditional methods is needed.

We thank the reviewer for pointing out the lack of clarity in the discussion of lines 5 and 6.

We’ve re-written that section of the discussion, and reformatted Figure 3 to enhance clarity.

We agree, a comparison to traditional HHBlits could be interesting, but we don’t expect to see stronger performance from the pLM-predicted profiles than from traditional HHBlits, just as we don’t see stronger performance from pLM-hmmscan or pLM-Foldseek than from the traditional variants. We think that the advantages of pLM based amino acid hmm searches are primarily speed. There are many variables that can influence speed of generating an MSA and HMM profile, but in general we expect that it will be much slower than generating an HMM profile from a pLM.

We don’t know why making profiles of 3Di sequences doesn’t improve search sensitivity, we just think it’s an interesting result that is worth presenting to the community. Perhaps someone can figure out how to make it work better.

Figure 3 and its associated text are hard to follow due to the abundance of colors and abbreviations used. One figure attempting to explain multiple distinct points adds to the confusion. Suggestion: Splitting the figure into two panels comparing (A) Foldseek-derived searches (lines 7-10) and (B) language-model derived searches (line 3-6) to traditional methods could enhance clarity. Different scatter markers could also help follow the plots more easily.

We thank the reviewer for this helpful comment. We’ve reformatted Figure 3 as suggested, and we think it is much easier to read now.

The justification for using Foldseek without amino acids (3Di-only mode) is not clear. Its utility should be described, or it should be omitted for clarity.

To us, the use of 3Di-only mode is of great theoretical interest. From our perspective, this is one of our most significant results. Previous methods, such as pLM-BLAST and related methods, have made use of very large positional embeddings to achieve sensitive remote homology search. We show that with the right embedding, you don’t need very many bits per position to get dramatically improved search sensitivity from Smith-Waterman, compared to amino acid searches. We also doubt that predicted 3Di sequences are the optimal small encoding for remote homology detection. This result and observation opens up an exciting avenue for future research in developing small, learned positional embeddings that are optimal for remote homology detection and amenable to SIMD-optimized pre-filtering and Smith-Waterman alignment steps.

We’ve expanded the discussion, explaining why we are excited about this result.

Figure 2 is not described, unclear what to read from it.

It's just showing that ESM-2-derived amino acid probabilities closely resemble amino acid frequencies in MSAs. We think it gives readers some visual intuition about why predicted profile HMMs perform as well as they do. We’ve added some additional explanation of it in the text.

**Recommendations for the authors:**

**Reviewer #1 (Recommendations For The Authors):**
The paper would mainly benefit from a more comprehensive benchmark:We suggest that the authors extend the benchmark by including the reference methods (HHpred and Foldseek) run with their original representations, i.e., MSAs obtained with 2-3 iterations of hhblits (for HHpred) and experimental or predicted structures (for Foldseek). HHpred profile-profile comparisons and Foldseek structure-structure comparisons would be important reference points for assessing the applicability of the proposed approach in distant homology detection. It is also essential to compare the method with other emerging tools such as EBA (DOI: 10.1101/2022.12.13.520313), pLM-BLAST (DOI: 10.1101/2022.11.24.517862), DEDAL (DOI: 10.1038/s41592-022-01700-2), etc.We also suggest using an evolutionary-oriented database for the benchmark, such as ECOD or CATH (these databases classify protein domains with known structures, which is important in the context of including Foldseek in the benchmark). We ran a cursory benchmark using the ECOD database and generated HH-suite .hhm files (using the single_seq_to_hmm.py and hhsearch_multiple.py scripts). Precision and recall appear to be significantly lower compared to "vanilla" hhsearch runs with MSA-derived profiles. It would also be interesting to see benchmarks for speed and alignment quality.The pLM-based methods for homology detection are an emerging field, and it would be important to evaluate them in the context of distinguishing between homology and analogy. In particular, the predicted Foldseek representations may be more likely to capture structural similarity than homology. This could be investigated, for example, using the ECOD classification (do structurally similar proteins from different homology groups produce significant matches?) and/or resources such as MALISAM that catalog examples of analogy.

We’ve added the SCOPe40 benchmark, which we think at least partially addresses these comments, adding a comparison to pLM-BLAST, ProstT5, and AF2 followed by Foldseek. The question of Analogy vs homology is an interesting one. It could be argued that the SCOPe40 benchmark addresses this in the difference between Superfamily (distant homology) and Fold (analogy, or very distant homology).

Our focus is on remote homology detection applications rather than alignment quality, so we don’t benchmark alignment quality, although we agree that those benchmarks would be interesting.

Page 2, lines 60-67. This paragraph would benefit from additional citations and explanations to support the superiority of the proposed approach. The fact that flattened embeddings are not suitable for annotating multidomain proteins seems obvious. Also, the claim that "current search implementations are slow compared to other methods" should be supported (tools such as EBA or pLM-BLAST have been shown to be faster than standard MSA-based methods). Also, as we mentioned in the main review, we believe that the generated pseudo-profiles and fine-tuned ESM2 predictions should not be called "smaller positional embeddings".

Discriminating subdomains was a major limitation of the influential and widely-cited PfamN paper (Bileschi et al., 2022), we’ve added a citation to that paper in that paragraph for readers interested in diving deeper.

To address the question of speed, we’ve included data preparation and search benchmarks as part of our presentation of the SCOPe40 benchmark.

Finally, we were not sure why exactly every 7th residue is masked in a single forward pass. Traditionally, pseudo-log likelihoods are generated by masking every single token and predicting probabilities from logits given the full context - e.g. https://arxiv.org/pdf/1910.14659.pdf. Since this procedure is crucial in the next steps of the pipeline, it would be important to either experiment with this hyperparameter or explain the logic used to choose the mask spacing.

We’ve added discussion of the masking distance to the Methods section.

**Reviewer #2 (Recommendations For The Authors):**
While the code and data for the benchmark are available, the generation of searchable databases using the methods described for a popular resource such as Pfam, AFDB, SCOP/CATH which can be used by the community would greatly boost the impact of this work.

3Di sequences predicted by ESM-2 3B 3Di can easily be used as queries against any Foldseek database, such as PDB, AFDB, etc. We’ve added Figure 4E to demonstrate this possibility, and added some related discussion.

Minor: In line 114, the text should likely read "compare lines 7 and 8" instead of "compare lines 6 and 7."

We’ve clarified the discussion of Figure 3.